# Nutrition, Physical Activity and Supplementation in Irritable Bowel Syndrome

**DOI:** 10.3390/nu15163662

**Published:** 2023-08-21

**Authors:** Marcelina Radziszewska, Joanna Smarkusz-Zarzecka, Lucyna Ostrowska

**Affiliations:** Department of Dietetics and Clinical Nutrition, Medical University of Bialystok, ul. Mieszka I 4B, 15-054 Bialystok, Poland; joanna.smarkusz-zarzecka@umb.edu.pl (J.S.-Z.); lucyna.ostrowska@umb.edu.pl (L.O.)

**Keywords:** IBS, irritable bowel syndrome, FODMAP, diet, nutrition, supplements, physical activity

## Abstract

Irritable Bowel Syndrome (IBS) is a chronic, recurrent functional disorder of the intestine diagnosed based on the Rome IV criteria. Individuals suffering from IBS often associate the severity of their symptoms with the food they consume, leading them to limit the variety of foods they eat and seek information that could help them determine the appropriate selection of dietary items. Clear nutritional recommendations have not been established thus far. NICE recommends a rational approach to nutrition and, if necessary, the short-term implementation of a low FODMAP diet. Currently, the FODMAP diet holds the greatest significance among IBS patients, although it does not yield positive results for everyone affected. Other unconventional diets adopted by individuals with IBS lack supporting research on their effectiveness and may additionally lead to a deterioration in nutritional status, as they often eliminate foods with high nutritional value. The role of physical activity also raises questions, as previous studies have shown its beneficial effects on the physical and mental well-being of every individual, and it can further help alleviate symptoms among people with IBS. Supplementation can be a supportive element in therapy. Attention is drawn to the use of probiotics, vitamin D, and psyllium husk/ispaghula. This review aims to analyze the existing scientific research to determine the impact of various food items, physical activity, and dietary supplementation with specific components through dietary supplements on the course of IBS.

## 1. Introduction

Irritable Bowel Syndrome (IBS) is one of the most commonly diagnosed chronic and recurrent functional gastrointestinal disorders. However, a definitive diagnostic standard for IBS has not been established [1]. Most commonly, the diagnosis is made according to the Rome IV criteria [2,3,4]. The primary diagnostic criteria for IBS include the presence of abdominal pain, bloating, constipation, and/or diarrhea without any morphological or biochemical changes [1,4]. According to the Rome IV criteria, in order to diagnose IBS, symptoms must persist for at least six months before making the diagnosis, and the diagnostic criteria must be met for the last three months [2]. Abdominal pain must occur at least one day per week in the past three months, be recurrent in nature, and meet at least two additional criteria, such as abdominal pain correlated with bowel movements, abdominal pain correlated with changes in bowel frequency, or abdominal pain correlated with changes in stool consistency [2,5]. If all these criteria are met and there are no additional alarming symptoms, there is no need for further diagnostic tests and the diagnoses of IBS can be set [1], with further characterization according to the Bristol Stool Form Scale that classifies IBS into four subtypes: Irritable Bowel Syndrome with Constipation (IBS-C), Irritable Bowel Syndrome with Diarrhea (IBS-D), Mixed Irritable Bowel Syndrome (IBS-M), and Unsubtyped Irritable Bowel Syndrome (IBS-U) [1,2,4,6].

Typical symptoms of IBS include abdominal pain and a change in bowel habits (diarrhea or constipation, often occurring alternately) [4,7]. Additional accompanying symptoms may include heartburn, indigestion, chest pain, bloating, urgency to have a bowel movement, feeling of incomplete evacuation, chronic pelvic pain, fibromyalgia, and migraines [1,4]. Due to the adverse impact of these symptoms on mental well-being, individuals with IBS are more likely to experience anxiety and depression [1,4,7]. The presenting symptoms vary in intensity and exhibit significant variability throughout the course of the disease [7,8].

The occurrence of IBS is characterized by significant variability both between countries and within a single country [8], affecting about 7–21% of the general population, most commonly reported in South American countries [3,4,8,9]. IBS is diagnosed twice as often in women compared to men [9], and although the condition can be diagnosed at any age, it is most commonly diagnosed in women before the age of 50 [4,8].

The etiopathogenesis of IBS is multifactorial and currently not fully understood [4]. According to existing analyses, it is possible that IBS results from interactions between genetic factors, psychiatric disorders, dysregulation of the hypothalamic-pituitary-adrenal axis, visceral hypersensitivity, and inflammatory states within the gastrointestinal tract as well as other parts of the body [1,3,5].

Treating patients with IBS is quite challenging due to the lack of a well-understood pathophysiology of the disease and the variability of symptom occurrence throughout its course. Pharmacological treatment focuses on symptom relief. Commonly used medications include those that regulate bowel movements (increasing the frequency of bowel movements for IBS-C or reducing frequency for IBS-D), analgesics, and antispasmodics [10]. Pharmacological treatment should be tailored individually based on the type and severity of the predominant symptoms of IBS [11].

However, the greatest value in the treatment process is attributed to appropriate nutrition and lifestyle changes. Making changes in these areas can have significant effects on improving the quality of life for individuals with IBS. Even up to 70% of individuals with IBS correlate the occurrence of their symptoms with their dietary habits [4,7,9]. The association between nutrition and the course of IBS leads to a reduction in the variety of foods consumed by patients. These dietary restrictions often result in inadequate intake of energy and nutrients, which can exacerbate symptoms and worsen IBS patients’ quality of life [12]. 

In addition to dietary changes, important modifiable factors for individuals with the condition that can influence the course of IBS are supplementation and physical activity. Dietary supplements, although their effects on the human body are not precisely proven, may have beneficial effects on human health [13]. Individuals with IBS can benefit from additional supplementation in terms of reducing symptoms, improving well-being, and enhancing their quality of life [13]. 

Due to the accompanying symptoms of the disease, patients often limit their engagement in physical activity. It has been shown that physical activity is essential for maintaining good physical and mental health and can also prevent gastrointestinal symptoms among healthy individuals [14]. Based on existing research, it is suspected that appropriately tailored exercise regimens may have positive effects on the course of the disease among individuals with IBS [14,15,16]. 

Due to the limited evidence regarding the impact of different food products, physical activity, and selected dietary supplementation on the course of IBS, inconclusive guidelines are available, and both patients and professional medical personnel lack reliable information regarding interventions that encompass rational dietary modifications, physical activity, and the inclusion of appropriate dietary supplements. Therefore, the aim of this study is to review the existing literature that describes the interventions effective for IBS.

## 2. Materials and Methods

A systematic literature search was conducted using the PubMed database to identify studies relevant to the current review. The following search string was used: (“irritable bowel syndrome” OR “IBS” OR “microbiome” OR “intestinal diseases” OR “colorectal cancer”) AND (“treatment” OR “diet” OR “low FODMAP diet” OR “FODMAP” OR “fiber” OR “vegetables” OR “fruits” OR “legumes” OR “grain products” OR “gluten-free diet” OR “gluten” OR “dairy products” OR “fermented dairy products” OR “lactose-free diet” OR “lactose” OR “fish” OR “omega-3 fatty acids” OR “eggs” OR “meat” OR “protein” OR “processed meat” OR “processed food” OR “supplementation” OR “probiotics” OR “psyllium” OR “vitamin D”). We attempted to limit the search to articles published within the last 5 years; however, if no data were available within this range, studies from earlier years were also included. Among the 2000 literature items found, 87 practical guidelines, reviews, and clinical studies were selected for article preparation. We only included studies in English to which we had full access. The inclusion criteria also included studies involving groups of more than 20 people, lasting longer than one day, including study and control groups, and conducted only with adults. Included studies used questionnaires, food diaries, and stool or blood screening. Exclusion criteria included lack of access to full texts, imprecise results and study designs, studies involving groups of less than 20 subjects, shorter than one day, not including a study and control group and involving children, or inadequate relevance to the topic. The conditions for inclusion and exclusion of articles are shown in Figure 1.

## 3. Food Choices

Studies have been conducted to investigate the impact of different diets on the course of IBS, including gluten-free, lactose-free, and high-fiber diets. However, these studies are not sufficient to confirm the effectiveness of any dietary intervention for all individuals with IBS [17]. The British Society of Gastroenterology (BSG) recommends that nutritional consultations should be a first-line therapeutic strategy [18]. According to the National Institute for Health and Care Excellence (NICE), basic dietary recommendations for patients with IBS should include regular meal consumption, avoiding meal skipping and large meals, drinking about 2 L (8 cups) of fluids per day (mainly water and herbal teas), limiting consumption of carbonated and alcoholic beverages, as well as reducing intake of caffeine, fats, insoluble dietary fiber (found in whole grains, cereal flakes, bran), resistant starch (present in processed and reheated foods), and gas-producing foods (including certain fruits) [19,20]. If these recommendations do not produce therapeutic effects, the next step is to implement a low FODMAP diet (Fermentable Oligosaccharides, Disaccharides, Monosaccharides, And Polyols), which involves eliminating foods rich in fermentable oligosaccharides, disaccharides, monosaccharides, and polyols [18]. Table 1 presents foods rich in specific types of FODMAPs [21].

Although the low-FODMAP diet currently has the highest number of studies due to its popularity among patients with IBS, it can contribute to nutritional deficiencies and a reduction in gut microbiota diversity due to its significant restrictions. Additionally, the evidence for its effectiveness is of very low quality, and further research on dietary treatment for individuals with IBS is necessary [17,22]. 

Few studies are also examining the feasibility of other diets in the course of IBS. To date, gluten-free, lactose-free and high-fiber diets are of most interest, in addition to the NICE indications and the low FODMAP diet. Clinical studies conducted to date show inconclusive results regarding the effectiveness of the aforementioned diets. It is worth noting that the improvement of symptoms accompanying IBS depends on the group of people studied and additional factors that may affect nutrient tolerance. It can be noted that a low-FODMAP diet and additional consumption of psyllium may be beneficial in the course of IBS. However, attention should be paid to the timing of the diets and dosages. On the other hand, gluten-free and lactose-free diets have positive effects only on a case-by-case basis depending on additional factors that interfere with nutrient tolerance [23,24,25,26,27,28,29,30]. The results of selected clinical trials are shown in Table 2.

Ingredients in elimonated and included products during these diets can affect the course of IBS. In a gluten-free diet, grain products that contain gluten are eliminated, but also other ingredients that affect the fermentation processes in the intestines and thus gastrointestinal symptoms [17,23,31]. Lactose, a milk sugar eliminated during a lactose-free diet, can also affect gastrointestinal processes [17,32]. On the other hand, the intake of dietary fiber, particularly water-soluble fiber, which influences the density, volume, consistency of stools and the composition and functioning of the intestinal microbiota, is increased in a high-fiber diet [17]. However, the results of the studies conducted so far are inconclusive and insufficient to recommend a single diet for every person with IBS [23,24,25,26,27,28,29,31,32]. The effectiveness of diets and their impact on the course of IBS is presented in Table 3.

It is worth noting that these diets may expose those following them to the elimination of valuable nutrients and the possibility of deficiencies [17,21,30,33]. This manuscript will explain and discuss that diets for IBS patients could come with deficiencies, and that specific iterative food choices for individual patients would be best to establish and consider more integrative diet and food group choices based on personal effectiveness.

## 4. Food Groups

### 4.1. Vegetables and Fruit

The World Health Organization (WHO) recommends daily consumption of 5 servings of fruits and vegetables [34]. These products provide biologically active substances (such as carotenoids, polyphenols, vitamin C, and others), are a primary source of many minerals (including potassium and magnesium), and contain dietary fiber. Due to their high nutritional value, fruits and vegetables can have a beneficial effect on human health, and reduce the risk and progression of many diseases, overall mortality, as well as disease-related mortality [34]. Individuals with IBS may restrict their consumption of vegetables and fruits due to the symptoms they experience. This is confirmed by a study conducted by Bohn et al., which showed that individuals with IBS reported intensified symptoms after consuming apples and plums [35]. Additionally, Monsbakken et al. observed that individuals with IBS tend to avoid consuming onions [36]. NICE guidelines limit the daily fruit intake for people with IBS to 3 servings (with each serving being approximately 80 g) [19,37]. Furthermore, they recommend eliminating onions, garlic, cabbage, artichokes, beans, peas, and watermelon from the diet. The FODMAP diet expands the list of allowed fruits and vegetables to include the white part of leeks, scallions, mushrooms, Brussels sprouts, asparagus, cauliflower, beets, fennel, as well as apples, pears, cherries, apricots, nectarines, plums, mango, lychee, longan, dried fruits, fruit juices, and canned fruits [38]. These recommendations are based on the composition and properties of these vegetables and fruits. They contain a high amount of fructose, fructans, and galactans, as well as water-insoluble dietary fiber. These compounds can cause increased production of methane and hydrogen, as well as short-chain fatty acids in the intestines, increase the influx of water into the intestinal lumen, and stimulate gastrointestinal peristalsis [38]. As a result, they can exacerbate symptoms occurring in IBS, such as bloating, abdominal pain, or diarrhea. However, it is worth noting that individuals with IBS can experience positive effects by incorporating into their diet products, including vegetables and fruits, that contain water-soluble dietary fiber.

Consuming dietary fiber can be challenging for individuals with IBS. According to NICE guidelines, these individuals should avoid consuming insoluble fiber (found, among others, in wheat bran). However, incorporating soluble fiber (found in plants like psyllium or oats) into their daily diet, even in the form of supplementation, may be beneficial [19,39]. It is worth noting that recommendations regarding dietary fiber intake are influenced by the type of IBS, as well as the type, amount, and source of fiber [13]. Insoluble dietary fiber has the ability to bind water and increase stool volume, thus stimulating intestinal peristalsis. However, existing studies indicate that this type of fiber does not contribute to the improvement of IBS symptoms and may even exacerbate symptoms in individuals with diarrhea-predominant IBS, leading to abdominal pain, diarrhea, gas, and bloating [39,40]. On the other hand, soluble fiber forms gels in the gastrointestinal tract, increasing stool bulk and improving its transit through the intestines [39]. It can also directly or indirectly positively influence the gut microbiota, improving its structure and function, as well as stimulating the growth of beneficial bacteria (such as *Lactobacillus* sp. and *Bifidobacteria* sp.) [37,41]. This type of fiber alleviates symptoms in individuals with IBS, reducing abdominal pain, bloating, and regulating bowel movements [13]. The positive effects of soluble dietary fiber can be observed in both diarrhea-predominant and constipation-predominant forms of IBS [13]. However, existing studies are insufficient to determine the recommended amount of fiber in the diet [42,43]. It is crucial to gradually introduce and increase the consumption of dietary fiber [41,43]. The American Academy of Nutrition and Dietetics recommends a daily intake of 25 g of dietary fiber for women and 38 g for men with IBS. Natural food sources such as vegetables, fruits, whole grains, legumes, nuts (almonds, hazelnuts), and seeds (pumpkin seeds, sunflower seeds) should be the primary sources of fiber [13,42]. However, achieving the recommended intake can be challenging, and in such cases, supplementation may be considered [42]. Ground flaxseed, consumed in an amount of up to 2 tablespoons per day, is a recommended form of dietary fiber enrichment [13].

Individuals with IBS should primarily consume vegetables and fruits low in insoluble dietary fiber and rich in soluble fiber fractions. When it comes to vegetables, they can choose carrots, tomatoes, zucchini, eggplant, cucumbers, bell peppers, broccoli, asparagus beans, spinach, pumpkin, or lettuce. Well-tolerated fruits include berries, raspberries, strawberries, honeydew melon, cantaloupe, grapes, oranges, and lemons. However, it is important to note that individuals with IBS should not completely eliminate vegetables and fruits that are contraindicated. Instead, they should individually assess their tolerance to specific types of vegetables and fruits. It is also important to consider not only the type of product but also the quantity and preparation methods. It is worth noting that the highest amount of fructans is found in the skin of vegetables and fruits, so patients may tolerate these products well after peeling them [44,45]. Individuals with IBS should aim to consume 3–5 servings of vegetables per day and 2–3 servings of fruits per day [13].

In summary, vegetables and fruits are crucial elements of every person’s diet as they provide a rich source of nutrients, especially vitamins and minerals. Individuals with IBS cannot omit them from their diet but need to pay attention to the type of products chosen, their quantity, and observe their individual reactions to them. Proper preparation of vegetables and fruits is also important. If raw consumption is poorly tolerated, trying cooked, baked, pureed, or juiced forms may be helpful. Removing skins can also be beneficial. Not only the consumption of vegetables and fruits but also the consumption of legume seeds can pose difficulties for patients.

### 4.2. Legume Seeds

Legume seeds are an excellent source of protein, minerals (including zinc, iron, magnesium, potassium), vitamins (including vitamin B group, such as vitamin B9), as well as biologically active compounds (such as phenolic acids, flavonoids), and dietary fiber. Due to their relatively high nutritional value of protein, they serve as an ideal alternative to animal-derived products, which is particularly important for individuals following vegetarian and vegan diets [46,47].

The recommendations for consuming legume seeds vary depending on the country. However, it is suggested that incorporating 100 g (half a cup or 125 mL) of cooked legume seeds into the daily diet can be beneficial for health due to their nutritional value [47]. However, the absorption and bioavailability of protein, minerals, and vitamins may be impaired by anti-nutritional factors, such as tannins or trypsin inhibitors. The presence of galactooligosaccharides may also contribute to gastrointestinal symptoms such as abdominal pain, bloating, and gas [47,48]. It’s worth noting that both the nutritional and anti-nutritional content varies depending on the type of legume seeds. Peas have the highest content of α-galactosides, while lentils have the lowest [48]. Proper technological processing can help reduce the levels of these unfavorable components. One of the most effective methods for reducing the water-soluble components, including oligosaccharides, is soaking legume seeds [48]. Reduction of oligosaccharides can also be achieved through extrusion, pressure cooking, or regular boiling [48,49]. Studies have shown that pressure cooking can lead to a reduction of these compounds in cooked seeds by 14–77% compared to raw seeds. Another good alternative for consuming legume seeds is incorporating pasta made from legume flour into the diet. It has also been observed that the combination of soaking and pressure cooking significantly increases the effectiveness of removing galactooligosaccharides [48].

As shown in a study conducted by Bohn et al., as many as 37% of the individuals with IBS reported experiencing symptoms after consuming beans and lentils [35]. Similarly, in the study by Monsbakken et al., 16% of the participants completely eliminated the consumption of legume seeds [36]. NICE guidelines recommend avoiding the consumption of beans. Additionally, the low FODMAP diet suggests the exclusion of lentils, beans, and peas due to their high content of glu-cans and fructans [38]. Soybeans, lentils, chickpeas, beans, peas, and fava beans can be classified within this group of food items.

In summary, patients with IBS should consider incorporating legume seeds into their daily diet, taking into account the type and processing methods. Due to their high digestibility and lower content of oligosaccharides, individuals with the condition can start by introducing small amounts of lentils, which may be better tolerated. However, proper processing is necessary. If good tolerance is observed, one can attempt to increase the quantity and introduce other types of legume seeds. It’s worth noting that oligosaccharides are not only found in legume seeds but also grains.

### 4.3. Grain Products

Grain products are the main source of energy and carbohydrates in the diet of every healthy individual [17]. Among individuals with IBS, consumption of products from this group may be associated with an exacerbation of symptoms. The use of a gluten-free diet has become common to alleviate the symptoms of the disease. A study conducted by Bohn et al. revealed that 24% of the surveyed individuals experienced symptom exacerbation after consuming flour [35]. In Monsbakken et al.’s analysis, 10% of the participants avoided consuming wheat flour [36]. Furthermore, Reuze et al. observed in their study that individuals with IBS more frequently than the control group partially or completely avoided gluten-containing products, and they more commonly reported gluten sensitivity or intolerance (although not all of these individuals had a medical diagnosis) [50]. The main reason for avoiding gluten was the perceived discomfort after its consumption, which was more frequently reported by individuals with IBS. After eliminating these products, individuals noted greater physical and psychological comfort [50]. NICE guidelines recommend avoiding wheat flour and products based on it. Additionally, the guidelines also recommend avoiding whole grain products, bran, and brown rice due to their high content of insoluble dietary fiber [19]. When following the low FODMAP diet, not only wheat products (bread, pasta, biscuits, cakes, etc.) should be excluded but also all barley and rye products [38]. However, it is worth noting that symptom relief occurs not after eliminating gluten but after eliminating wheat consumption [51]. This suggests that symptom exacerbation after consuming wheat products is not related to the presence of gluten as previously believed, but rather caused by other components of wheat. High levels of gluten, fructans, amylase and trypsin inhibitors, as well as lectins, can all impact the course of the disease [17,50,51]. All these components can contribute to bloating, abdominal pain, and diarrhea [17].

Therefore, individuals with IBS should replace wheat with spelt products, as recommended by NICE guidelines [38]. Daily diet can also include oats, rice, quinoa, or corn [13]. People with IBS can utilize gluten-free products, which are mainly made from rice and/or corn [51]. Spelt bread contains only 0.14 g of fructans per 100 g of the product, while gluten-free bread contains 0.19 g/100 g [51]. NICE recommends individuals experiencing gas and bloating consume oatmeal or flaxseed, which can be beneficial in alleviating symptoms [19,37]. The beneficial effect of flaxseed is due to its content of soluble dietary fiber. In the case of IBS-C, flaxseed helps relieve constipation, abdominal pain, and bloating. It is recommended to include up to 2 tablespoons (6–24 g/day) of ground flaxseeds in the daily diet, consumed with fluids (150 mL of fluids/1 tablespoon of ground flaxseed) [43,52]. Grain products should be consumed up to 6 servings per day [13].

In summary, a gluten-free diet should not be applied to all patients with IBS. Each individual should observe whether symptoms occur after consuming wheat products and determine their own tolerance. It is also crucial to identify the quantity of these products that triggers discomfort. In the case of poor tolerance, it is not necessary to eliminate grain products altogether but rather to opt for alternative grains. Specifically, spelt is highlighted due to its high nutritional value and good tolerance.

### 4.4. Milk and Dairy Products

Apart from a gluten-free diet, a lactose-free diet also generates significant controversy. Lactose is a disaccharide found in milk and dairy products [43]. According to NICE guidelines and the principles of the low FODMAP diet, the consumption of milk and dairy products should be avoided [38]. Lactose in the human digestive system is broken down into glucose and galactose by an enzyme called lactase, which is secreted in the initial segment of the small intestine. After lactose is broken down, it is absorbed and does not reach the further segments of the intestines [53]. As a result, it does not contribute to the occurrence of diarrhea, bloating, or gas. However, if an individual has a deficiency or absence of lactase enzyme secretion, lactose passes into the large intestine, where it is fermented by the intestinal microbiota. This fermentation produces hydrogen and short-chain fatty acids, which can lead to gastrointestinal symptoms [17,43]. In Monsbakken’s study, as many as 35% of individuals declared avoiding the consumption of milk, and in Bohn’s analysis, nearly 50% of people reported experiencing symptoms after consuming milk and its products [35,36]. Studies show that individuals with IBS often report milk product intolerance, but this is not reflected in lactose absorption assessments (hydrogen breath test) [13,43]. Therefore, lactose intolerance is rather rare among affected individuals, and eliminating milk and dairy products may be unwarranted. According to the British Dietetic Association, which summarized previous studies on the link between lactose intolerance and IBS, it is important to conduct assessments evaluating lactose tolerance. However, there is no indication to recommend a low-lactose or lactose-free diet to every patient [52].

Milk and dairy products are an important part of everyone’s diet as they provide a source of protein, essential minerals (particularly calcium), and vitamins. Following a dairy-free diet can result in calcium deficiencies in the diet [43,54]. Individuals with IBS should consume 2–3 servings of milk and dairy products per day (1 serving being 200–250 mL of milk or 200–250 g of yogurt or 80–100 g of fresh cheese or 30–50 g of hard cheese) [13,43]. It’s important to note that even in the presence of lactose intolerance, it is not necessary to completely eliminate it from the diet [54]. It has been observed that a significant portion of individuals with documented lactose intolerance can tolerate daily lactose consumption of up to 12–15 g [54]. Therefore, these individuals should rather reduce their intake of lactose-containing products rather than eliminate them. Including at least a small amount of dairy products has benefits as it allows for better digestion, bioavailability, and replenishment of the nutrients they contain. Patients can gradually incorporate milk, starting with small amounts (about 30–60 mL/day), preferably consumed with other foods and not on an empty stomach. Mature cheeses are usually well-tolerated as they have significantly lower amounts of lactose (0.1–0.9 g of lactose per 30 g of cheese) [54]. Additionally, individuals with lactose intolerance can replace milk and dairy products with lactose-free or plant-based alternatives. In this case, they can consume lactose-free milk, yogurts, plant-based beverages, and other products (such as rice and almond milk) [13]. However, it’s important to read labels and choose products that are fortified with calcium and vitamins (including B2, B12, D, A) [54].

To date, there is an insufficient number of studies assessing the impact of fermented dairy products on the course of IBS. Fermented dairy products contain lactic acid bacteria, including *Streptococcus thermophilus* and *Lactobacillus delbrueckii* subsp. *Bulgaricus*. Additional bacteria from the *Lactobacillus* and *Bifidobacterium* genera are often added to fermented products, enabling them to fulfill additional probiotic functions. The microorganisms present in these products must be live, and their quantity should be greater than 10^7^ CFU/g [55]. A review conducted by Savaiano et al. in 2021 analyzed seven randomized clinical trials and demonstrated that the consumption of yogurt and kefir had a beneficial effect on lactose digestion and contributed to increased lactose tolerance in individuals with diagnosed lactose malabsorption [55]. This review also included five studies evaluating the impact of fermented dairy products on gastrointestinal symptoms accompanying IBS. The included studies showed that the consumption of fermented dairy products alleviated symptoms occurring in the course of the disease, and the effects were more favorable compared to the consumption of non-fermented dairy products [55]. Additionally, it was observed that the consumption of fermented dairy products reduced the risk of colorectal cancer [55]. Based on the conducted review, it can be concluded that individuals with IBS should not refrain from consuming fermented dairy products. Often, individuals with lactose intolerance eliminate their consumption, which is unjustified as these products are often well-tolerated even in the presence of impaired disaccharide digestion. Fermented dairy products, due to their composition, not only do not induce discomfort but may also contribute to reducing the experienced symptoms.

In summary, individuals with IBS are not recommended to follow a lactose-free diet. Each patient should assess their own tolerance to milk and dairy products, both in terms of the type of product and quantity. In case of observing an exacerbation of symptoms, regular milk and its derivatives can be replaced with lactose-free or plant-based products. It is also important not to forget about fermented dairy products, which can have a positive impact on the experienced symptoms during the course of the disease. Before completely eliminating dairy, attempts should be made to assess the tolerance to different types of dairy products and their quantities. If a patient reduces or completely eliminates the consumption of milk and dairy products, they may be at risk of protein deficiency in their diet. In such cases, the patient must remember to include other foods rich in high-quality protein in their meals, such as eggs, meat, or fish.

### 4.5. Meat, Fish, Eggs

So far, there is a lack of studies regarding the impact of consuming meat, fish, and eggs on the course of IBS. These products are a primary source of high-quality protein for every individual. Meat is a major source of vitamin B12 and heme iron [56]. Fish is a primary source of polyunsaturated omega-3 fatty acids and contains vitamin D [57]. Eggs provide a small amount of energy (approximately 140 kcal/100 g) but are rich in vitamins, particularly from the B group, as well as minerals such as iron, zinc, and calcium [58]. It is recommended that individuals with IBS consume 2–3 servings of meat, fish, or eggs per day (1 serving is approximately 100–125 g of meat, 125–150 g of fish, or 60–80 g of eggs) [43].

Among meat products, it is important to pay attention to the consumption of red meat and processed meat products. According to the International Agency for Research on Cancer (WHO-IARC), processed meat products have been classified as “carcinogenic to humans”, while red meat is classified as “probably carcinogenic to humans”. These products contain nitrates, nitrites, as well as heterocyclic amines and polycyclic aromatic hydrocarbons [59,60]. These compounds, through their influence on the mucous membrane of the colon, increase the likelihood of developing cancer. They also contain significant amounts of heme iron, which, by stimulating the formation of carcinogenic N-nitroso compounds in the lumen of the intestine, likely promotes the development of tumors in the gastrointestinal tract [59]. Additionally, processed meat products may stimulate tumorigenesis because they contain a much higher amount of fat compared to red meat, which significantly increases the synthesis of secondary bile acids. However, this mechanism is not precisely understood in humans [59].

A review of the results of the multicenter prospective cohort study called the European Prospective Investigation into Cancer and Nutrition (EPIC) was conducted [61]. The review included a total of 110 high-quality studies that examined the impact of various food groups on colorectal, lung, breast, and prostate cancer. It was demonstrated that higher consumption of fish, along with a simultaneous reduction in the intake of red meat and processed meat products, was associated with a decreased risk of developing colorectal cancer [61].

A meta-analysis conducted by Bolte et al. aimed to observe the influence of individual dietary components and food groups on the composition of the gut microbiota, as well as the development of intestinal inflammation and various disease conditions [62]. The meta-analysis included studies that assessed the impact of dietary habits, specific food products, and individual dietary components on the gut microbiota composition in four groups of individuals: those with Irritable Bowel Syndrome, Crohn’s Disease, Ulcerative Colitis, and a population of healthy individuals. It was shown that dietary strategies involving increased consumption of fish, nuts, legume seeds, and bread were associated with a reduction in the abundance of opportunistic microorganisms responsible for the synthesis of pro-inflammatory factors and endotoxins. Additionally, such dietary patterns contributed to increased colonization of the gut by bacteria such as *Roseburia*, *Faecalibacterium*, and *Eubacterium* spp. These microorganisms ferment dietary fiber into short-chain fatty acids, thus exhibiting anti-inflammatory effects. Furthermore, high fish consumption was associated with the colonization of the gut by *Roseburia hominis* and *Faecalibacterium prausnitzii*. The omega-3 fatty acids present in these products led to a decrease in the abundance of pathobionts and pro-inflammatory factors, while an increase in anti-inflammatory factors was observed. It was also demonstrated that high consumption of fast food and processed meat correlated with an increase in the abundance of *Ruminococcus gnavus, Akkermansia muciniphila*, and *Proteobacteria*, which influenced increased gut permeability and the development of intestinal mucosal inflammation. The study authors noted that well-designed dietary interventions characterized by increased consumption of fish, legume seeds, nuts, vegetables, and fruits, as well as low-fat fermented dairy products, while limiting the consumption of alcoholic beverages, high-fat and processed meat, and sweetened beverages, would impact the gut microbiome, thereby preventing and alleviating the course of intestinal mucosal inflammation [62].

In summary, patients should include meat, fish, and eggs in their diet. They should pay attention to the type of meat products they choose. The diet should include lean poultry (turkey, chicken), while the consumption of fatty meats (pork, goose, duck) and red meat should be limited. Processed meat products should be eliminated from the diet. The most important aspect is to increase the consumption of fish, especially fatty fish (salmon, herring), which, due to their high content of omega-3 fatty acids, may have positive effects on the course of IBS. Additionally, the inclusion of eggs in the diet should not be overlooked. Despite the lack of studies on the impact of eggs on IBS, they are a valuable source of protein, vitamins, and minerals, which can effectively supplement the insufficient intake of these nutrients from other food groups.

After analyzing all food groups and the recommended dietary guidelines by NICE (National Institute for Health and Care Excellence), it can be observed that the dietary approach for individuals with IBS varies depending on the specific type of the condition and should be individually tailored based on the course of the disease and symptoms. Figure 2 presents a summary of the most relevant dietary recommendations supported by previous research for individuals with IBS.

## 5. Supplementation

### 5.1. Probiotics

Probiotics are live microorganisms that, when taken in specific amounts, can have a positive impact on human health [20]. Probiotic preparations can be used in the treatment of IBS as they can contribute to changes in the composition of gut microbiota. Among individuals with IBS, abnormalities in the composition and functioning of gut-colonizing microorganisms are often observed, which may contribute to the onset of symptoms in the course of the disease [20].

A study was conducted that focused on comparing the composition of gut microbiota in individuals with IBS (study group: *n* = 24) and healthy individuals (control group: *n* = 23) [63]. Stool samples were collected from the study participants, and microbial genome analyses were performed. Individuals with IBS participating in the study exhibited a significantly different composition of gut microbiota compared to healthy individuals. It was observed that the microbiota of individuals in the study group was significantly depleted in the microorganisms *Coprococcus eutactus*, *Clostridium cocleatum*, and *Collinsella aerofaciens*. This analysis drew the researchers’ attention to the potential use of probiotic preparations for alleviating gastrointestinal symptoms occurring in the course of IBS [63].

One of the most commonly analyzed probiotic strains in IBS is *Lactobacillus plantarum 299v* [64]. These microorganisms are not pathogenic, they are resistant to the digestive enzymes of the gastrointestinal tract, and once they reach the intestines, they colonize the colonic mucosa. Through colonization of the large intestine and antimicrobial activity, mainly against pathogenic Gram-negative bacteria, this strain alleviates the inflammatory process taking place in the intestines. It also shows immunomodulatory effects as it influences the synthesis and secretion of pro-inflammatory and anti-inflammatory cytokines. *Lactobacillus plantarum 299v* also increases the production of carboxylic acid, acetic acid and propionic acid, thereby lowering the hydrogen potential in the colon, which supports the control of microbial proliferation in the colon [64]. A randomized controlled trial conducted in 2022 by Moeen-Ul-Haq et al. showed that the effect of *Lactobacillus plantarum 299v* administration on the course of IBS was not significantly different from placebo [64]. The study involved 190 individuals diagnosed with IBS. The subjects were divided into two equal groups, with one group receiving 5 × 1010 CFU of *Lactobacillus plantarum 299v* (study group) and the other receiving placebo (control group). After a four-week follow-up, an alleviation of abdominal pain, bloating and the sensation of incomplete bowel movements was noted; however, no significant differences were observed between the groups. The authors, therefore, concluded that supplementation with *Lactobacillus plantarum 299v* did not show significant efficacy in the treatment of IBS [64]. In contrast, a randomized experimental study also conducted in 2022 by Bednarska et al. noted that *Lactobacillus plantarum 299v* had a more beneficial effect on the course of IBS compared to placebo [65]. In this analysis, 30 people diagnosed with moderate IBS were assigned to one of two groups. The first group received an enema containing the bacterial strain (study group), while the second group received a placebo (control group). In both groups, the formulations were administered twice a day for a period of 14 days, and a biopsy from the distal part of the colon was taken from each participant before the start and end of the study, as well as questionnaires on IBS symptoms. The study noted that there was a significant reduction in intestinal mucosal permeability and improvement in transepithelial resistance (TER) in the study group relative to the control group. However, the authors show that confirmation of their findings requires further research [65].

An interventional, randomized, controlled clinical trial conducted by Gupta et al. analyzed the safety and efficacy of using the *Bacillus coagulans* LBSC (DSM17654) strain among individuals with IBS [66]. The study involved a total of 40 participants of both sexes with IBS, who were divided into two groups (20 individuals in each). The first group received preparations containing the bacterial strain at a dose of 6 billion CFU (colony-forming units) per day (2 billion CFU in powder form three times a day) for a duration of 80 days. The second group (control group) received placebo preparations (the same form of administration containing maltodextrins) for the same duration of time. The study authors demonstrated that *Bacillus coagulans* LBSC (DSM17654) had a beneficial impact on reducing gastrointestinal symptoms such as bloating, abdominal pain, diarrhea, constipation, nausea, vomiting, and also alleviated headache pain. Additionally, it was proven that the use of this probiotic strain, by alleviating the symptoms of the condition, can positively affect the well-being and mood of patients, as well as improve their quality of life [66].

The beneficial impact of probiotic use on the quality of life in individuals with IBS is also confirmed by a review conducted by Le Morvan de Sequeira et al. [67]. The authors analyzed 35 studies involving a total of 4717 individuals. The studies utilized single-strain or multi-strain probiotics (consisting of two to eight different strains) in the form of sachets, capsules, tablets, or liquids. The participants in the studies consumed probiotics ranging from 1 × 10^8^ to 9 × 10^11^ CFU (colony-forming units) per day (median = 1 × 10^10^ CFU/day). The intervention groups (using probiotics) were compared to placebo groups. After analyzing the results, the authors of the review concluded that the use of probiotics in the management of IBS can improve the quality of life of patients compared to placebo. The improvement in quality of life was likely associated with the impact of probiotics in alleviating somatic symptoms. Additionally, it was noted that *Bifidobacterium longum* NCC3001 could potentially be used as a psychobiotic as it reduces the reactivity of the limbic system, contributing to the alleviation of anxiety. However, this requires further confirmation in future studies [67].

A systematic review and meta-analysis of seventeen randomized controlled trials conducted by Wen et al. demonstrated that the use of probiotics can have a beneficial effect on intestinal peristalsis, stool frequency, and consistency [68]. The meta-analysis included studies conducted among adults with IBS-C (constipation-predominant IBS). In the randomized controlled trials, probiotic therapy was compared to placebo or to individuals not using any supplements. However, the meta-analysis did not consider the specific species, strains, doses, and regimens of probiotic use. The researchers observed that individuals using probiotics did not report any adverse effects. Compared to placebo, the use of probiotics resulted in accelerated intestinal transit, significantly increased stool frequency, and notably improved stool consistency [68].

The British Society of Gastroenterology also conducted a meta-analysis of 45 studies evaluating the use of different types of probiotics in the management of IBS [18]. The team members indicate that a significant beneficial impact on overall symptoms and abdominal pain among patients can be achieved through a combination of different bacterial strains in probiotic preparations. Less significant, yet positive therapeutic effects may also be obtained with the use of preparations containing individual strains of bacteria from the *Lactobacillus, Bifidobacterium*, and *Escherichia* genera [18]. 

The mechanism of action of probiotic preparations is not fully understood. It is noted that it may inhibit the development of IBS in particular post-infectious IBS, as they exhibit anti-inflammatory, antimicrobial and antiviral effects [69]. It is likely that probiotics modify the body’s immune system response by modifying the response of dendritic cells and stimulating the production of antimicrobial proteins [69,70]. Probiotics can relieve abdominal pain by increasing the expression of cannabinoid and opioid receptors [71]. Short-chain fatty acids, formed by probiotic-mediated fermentation in the gut, are a nutrient for intestinal epithelial cells resulting in improved integrity and enhanced intestinal mucosal barrier function. SCFAs are also responsible for lowering the pH of the gastrointestinal lumen, as a result of which they inhibit the growth of pathogenic bacteria [69]. However, it is important to note that the authors of all the studies conducted so far emphasize the need for more detailed randomized controlled trials that will further define the duration of use and dosage (CFU), as well as the species and strains of probiotics used [18,68].

In summary, therapies involving probiotic preparations can yield significant therapeutic effects in the management of IBS. Probiotics, by modifying the gut microbiota, can have a beneficial impact on gastrointestinal symptoms and the functioning of the digestive tract, thereby improving the well-being of individuals with IBS, their quality of life, and their experience of anxiety. However, the current body of research is insufficient to provide clear recommendations for IBS patients regarding the specific species and strains of probiotics to be used, as well as the duration of use and the dosage (CFU).

### 5.2. Psyllium Husk

Psyllium husk, also known as *Plantago ovata*, is the seed of the *Plantago ovata* plant. These seeds contain arabinoxylan, a highly branched polysaccharide that forms a gel-rich polymer abundant in arabinose and xylose. It is the main component of dietary fiber and is not digested by human digestive enzymes. However, once it reaches the large intestine along with food content, it is utilized by certain microorganisms colonizing the gastrointestinal tract. The selective utilization of arabinoxylan by the microbiota can result in changes in its composition, as demonstrated by Jalanka et al. in 2019 [72]. The researchers conducted two independent, randomized, controlled, placebo intervention studies. One study aimed to investigate the impact of psyllium husk supplementation on the microbiota composition of healthy individuals, while the other evaluated the same intervention among individuals with constipation. Significant modifications in gut functioning were observed in both groups, including changes in stool water content, transit time, and production of short-chain fatty acids (SCFA). The analyses also revealed slight alterations in the gut microbiota composition of the healthy group following supplementation, whereas individuals with constipation experienced significant changes in the composition of colon-residing microorganisms, notably a significant increase in the abundance of bacteria from the *Lachnospira* genus. Some strains of *Lachnospira* have the ability to produce lactate and acetate, which are further transformed into butyrate and propionate. Insufficient butyrate production reduces mucin secretion, which can lead to constipation. This study demonstrated that psyllium husk supplementation contributes to an increase in the abundance of gut microbiota responsible for SCFA production, indicating the positive effects of using such preparations [72].

The mechanisms of action of psyllium are currently not well understood. However, a 2023 study by Bretin et al. showed that psyllium can alleviate colitis by affecting bile acid metabolism [73]. The study was conducted among mice that were fed diets enriched with dietary fiber from various sources. It was noted that individuals fed psyllium were significantly better protected against colitis. It is likely that this relationship was due to induced changes in the composition of the intestinal microbiota and, most importantly, the ability of psyllium to alter bile acid production. Psyllium shows the ability to modify the synthesis of bile acids and increase their concentration in blood serum, resulting in the activation of the farnesoid X receptor, which in turn is involved in protection against colitis. The authors of the analysis note that it is worth focusing on further investigation of the mechanisms of action of psyllium [73].

Psyllium husk, also known as *Plantago ovata*, does not irritate the colon mucosa. It has a high water-holding capacity and exhibits dualistic normalizing activity on bowel movements. In cases of constipation, it softens hard stool, while in cases of diarrhea, it solidifies loose stool and stabilizes the frequency and form of bowel movements in individuals with IBS [74]. However, in order for psyllium husk to fulfill its functions, it is important to ensure an adequate intake of water along with the preparation. It has been noted that there is a lack of precise recommendations regarding the amount of water consumed, and additionally, the recommended dosage of psyllium husk is often too small (7–14 g/day) to fully fulfill its functions. It has been shown that the average fiber intake of adults in the United States is below 15 g per day, while the average daily fiber requirement for adults is around 25–38 g. Therefore, researchers suggest that at least 20 g of psyllium husk should be consumed along with a minimum of 500 mL of water [74].

The Canadian Association of Gastroenterology conducted a literature review to establish guidelines for managing IBS [75]. Specialists analyzed 15 randomized controlled clinical trials and performed a meta-analysis. The existing studies have shown that supplementation with psyllium husk yields positive effects in alleviating IBS symptoms compared to the use of a placebo or no treatment. It has been observed that the use of psyllium husk has a beneficial impact on the occurrence of constipation and diarrhea, and its mechanism of action involves changes in stool consistency, modifications in the production of fermentation products by the gut microbiota, and alterations in the composition of intestinal microorganisms. Therefore, the association recommends the use of psyllium husk supplementation among individuals with IBS as it is a cost-effective, safe, and patient-accepted form of treatment [75].

In summary, considering the effects of psyllium husk and the guidelines provided by the Canadian Association of Gastroenterology, the use of preparations containing this ingredient should be recommended among patients with IBS. According to the existing research, its utilization is safe and yields positive effects in treating IBS symptoms. However, when recommending the use of psyllium husk, it is important to remind patients to increase their fluid intake (at least 2 L per day) as it is crucial for the proper functioning of psyllium husk.

### 5.3. Vitamin D

Vitamin D is a fat-soluble vitamin that is converted into 25-hydroxyvitamin D and other compounds (such as 1.25(OH)2D) in the human body. The proper level of vitamin D in the body and the occurrence of deficiencies largely depend on geographic location and dietary habits. Vitamin D can be synthesized in the skin through exposure to sunlight or obtained through food, although dietary intake is usually very low [76]. Vitamin D is crucial for maintaining proper bone mineralization, reducing the risk of rickets in children and osteomalacia in adults, and minimizing the risk of fractures. However, vitamin D also plays many extra-skeletal functions, reducing symptoms in psoriasis, potentially influencing the course and alleviating symptoms of inflammatory bowel diseases, playing a role in immune system support, impacting autoimmune diseases, and current research suggests its potential in reducing the risk of certain cancers [76,77]. To prevent diseases and ensure the proper functioning of the skeletal system and other organs and systems in the human body, attention should be paid to vitamin D supplementation. According to current guidelines, there is no single recommended dose for vitamin D supplementation, and it should be adjusted individually for each person, but typically ranges from 800 to 2000 IU/day [78]. It is considered that the concentration of 25(OH)D in blood serum should not be lower than 50 nmol/L to prevent adverse effects of vitamin D deficiency [76]. Due to the fact that 20% of vitamin D is obtained through diet, gastrointestinal disorders have been observed as a possible cause of insufficient vitamin D levels in the blood serum. Therefore, it has been noted that vitamin D deficiency is more common among individuals with IBS than in the general population [77,79].

A study conducted in 2021 by Linsalata et al. aimed to assess the relationship between vitamin D serum levels, intestinal barrier structure, and symptoms among individuals with diarrhea-predominant Irritable Bowel Syndrome (IBS-D) [77]. The study ultimately involved 36 participants with IBS-D, among whom 44% had a vitamin D deficiency (<20 ng/mL), while the remaining portion of the group had low but within-normal-range vitamin D levels (≥20 ng/mL) in their blood serum. During the study, the participants attended three visits. The first visit involved providing basic information about the project and conducting all required tests. One week after the first visit, the patients had their second meeting, during which anthropometric analyses were performed, and individuals qualifying for intervention were identified. The final visit took place 12 weeks after implementing the recommendations from the first visit. During the third visit, all necessary examinations and analyses of dietary journals were conducted. The researchers noted that after 12 weeks of observation, the vitamin D serum levels significantly increased in both groups, particularly among individuals with a pre-existing vitamin D deficiency before the implementation of the new dietary recommendations. It was observed that individuals with a vitamin D deficiency prior to the intervention exhibited significantly more severe symptoms compared to those with normal vitamin D levels in their blood serum. However, after 12 weeks of following the recommendations, a substantial improvement in IBS symptoms was observed in both groups, with particular attention given to the regulation of bowel movements. The study also included analyses of intestinal permeability and the intensity of inflammation. It was demonstrated that an increase in vitamin D serum levels led to an improvement in the intestinal barrier in both groups and a reduction in pro-inflammatory factors (IL-6, IL-8). The researchers provided evidence that vitamin D serum levels play a crucial role in maintaining proper intestinal barrier function and influence the development of IBS. Vitamin D deficiency contributed to the heightened severity of disease-specific symptoms, central nervous system sensitivity, and symptoms of depression and anxiety [77].

A review conducted in 2022 by Huang et al. assessed whether the inclusion of vitamin D supplements could impact the treatment of IBS and positively influence the quality of life for affected individuals [80]. Among 149 studies, the researchers included four randomized controlled trials in their meta-analysis, involving a total of 334 participants, with 169 individuals with IBS receiving vitamin D supplementation and 166 individuals receiving a placebo. The duration and dosage of vitamin D supplementation varied across the publications [80]. The conducted review demonstrated that the use of vitamin D supplements can alleviate most symptoms associated with IBS (such as abdominal pain, diarrhea, constipation, and bloating) and positively impact the improvement of quality of life (including reducing the occurrence of depression and anxiety). The mechanism of action of vitamin D is not fully understood, but it is suggested that it reduces inflammation of the intestinal mucosa and alleviates psychological and psychiatric conditions [80].

A meta-analysis conducted in 2023 by Yan et al. indicates that vitamin D supplementation can alleviate gastrointestinal discomfort associated with IBS, possibly by increasing vitamin D receptor expression and, as a result, attenuating inflammatory responses of the intestinal mucosa. However, researchers indicate that the pathophysiology of the development of IBS is highly complex, making the effects of this vitamin on gastrointestinal motility, visceral hypersensitivity, gut-brain axis function, stress sensation and other factors affecting disease development likely to be much more complicated [81].

In summary, vitamin D is incredibly important for every individual. Due to limited synthesis in the skin and very low dietary intake, considering vitamin D supplementation is recommended for everyone. The recommended preventive dosage ranges from 800–2000 IU per day. It is particularly important to pay attention to the occurrence of vitamin D deficiencies among patients with IBS and to recommend vitamin D supplementation in this group, as the inclusion of supplements can have positive effects on the course of the disease. Supplementation can alleviate IBS symptoms and improve quality of life.

## 6. Physical Activity

Regular physical activity brings health benefits to the cardiovascular system and overall body functioning, and reduces the risk of many chronic diseases (including cardiovascular diseases, diabetes, and cancer), as well as overall mortality. It positively affects growth and development. Exercise not only influences the physical aspects of human life but also has mental benefits, improving well-being, enhancing cognitive processes such as thinking, learning, and judgment, and reducing symptoms of anxiety and depression [82]. According to the recommendations of the World Health Organization (WHO), every adult should engage in aerobic and moderate-intensity exercise for at least 150–300 min per week or at least 75–150 min per week of aerobic exercise at high intensity or a proportionate mix of both types of exercise [82]. Additionally, adults should engage in muscle-strengthening activities of moderate or high intensity at least two times per week to gain additional health benefits, activating all major muscle groups [82].

According to the guidelines of the National Institute for Health and Care Excellence, patient education regarding engaging in physical activity is necessary for the prevention and treatment of IBS [19]. Individuals with the condition should be provided with brief instructions that take into account the recommended exercises and are tailored to their accompanying symptoms. Short advice on physical activity aims to increase the motivation of individuals with IBS to engage in it more frequently. Particularly, these brief recommendations regarding physical activity are significant among patients with IBS who have previously had a relatively low level of physical activity [19].

To date, there have been limited and relatively low-quality studies that have analyzed the impact of physical activity on the course and treatment of IBS. In 2018, Sadeghian et al., as part of the SEPAHAN project (The Study on the Epidemiology of Psycho-Alimentary Health and Nutrition), conducted a cross-sectional study among nearly 5000 adult residents (both women and men) of Iran [15]. The study authors observed that individuals leading a sedentary lifestyle have a 27% higher risk of developing IBS compared to physically active individuals. It was also noted that physically active individuals exhibit better dietary habits. They consume more water, prioritize breakfast, maintain regular meal patterns, and chew their food thoroughly. However, the authors acknowledge that the mechanism by which physical activity influences the development of IBS remains unknown, and they emphasize the need for further prospective studies [15].

A randomized controlled trial was conducted among patients with IBS by Daley et al. [83]. In this study, individuals were divided into two groups: one group had access to basic medical care only, while the other group received additional advice on engaging in physical activity. Throughout the study, the quality of life and accompanying symptoms of IBS were analyzed, with particular emphasis on abdominal pain, diarrhea, constipation, and general symptoms. After a 12-week observation period, researchers observed significant differences between the groups regarding the accompanying symptoms of the disease. It was demonstrated that physical activity can have a significant impact on symptom relief, particularly in reducing the occurrence of constipation. Furthermore, no significant differences were observed between the groups in terms of improvements in quality of life [83].

Two controlled studies were also conducted in Sweden to investigate the impact of physical activity on the treatment process of IBS [84,85]. In the first study, patients were subjected to a 12-month observation period, while the median duration in the second study was 5.2 years. Researchers observed that engaging in physical activity of moderate to high intensity (such as walking, cycling, aerobics) three to five times a week, ranging from 20 to 60 min, had beneficial effects on gastrointestinal symptoms and the well-being of individuals with IBS. It was shown that compared to the control group, individuals in the intervention group reported better quality of life and reduced experience of depression and anxiety [84,85].

A study conducted in 2016 by Shahabi et al. [86] also confirmed the beneficial effects of physical activity among individuals with IBS. The study examined the impact of regular walking and practicing yoga on IBS symptoms and the well-being of the patients. The analysis demonstrated that individuals practicing yoga reported a reduction in somatic symptoms, while regular walking showed a tendency to alleviate gastrointestinal discomfort and reduce anxiety and negative emotions. However, it is worth noting that the study found a greater long-term impact of walking due to its higher regularity [86].

In summary, physical activity should be an important component of everyone’s lifestyle as it positively impacts overall health and body functioning, helps regulate stress, improves well-being and mood, and likely has positive effects on the severity and frequency of somatic symptoms. However, for patients with IBS, gentle, slow, and low-intensity activities such as walking, yoga, cycling, swimming, and aerobics are recommended. It is also necessary to conduct more research to determine which activities are most beneficial and have an impact on the well-being and quality of life of patients with IBS. Nevertheless, despite the limited amount of research available, individuals with IBS should be encouraged to engage in regular physical activity.

## 7. Conclusions

The first intervention in terms of nutrition among individuals with IBS should be basic dietary recommendations recommended by NICE, based on principles of balanced nutrition. The second step, in the absence of therapeutic effects, is to consider the use of a low FODMAP diet. However, symptoms and the patient’s nutritional status should be monitored, as this diet does not always reduce symptom severity and improve the quality of life for patients. Prolonged use without specialist supervision may lead to deterioration in the patient’s nutritional status.Other unconventional diets, such as lactose-free and gluten-free diets, are not recommended for individuals with IBS, as there is no clear evidence regarding their effectiveness. Moreover, they may lead to a deterioration in nutritional status and overall health in individuals who follow them.Due to concerns about experiencing symptoms related to consuming specific foods, individuals with IBS often unjustifiably eliminate those foods from their diet, putting themselves at a high risk of nutritional deficiencies.All food groups should be included in the diet of individuals with IBS. However, it is important to individually adjust the quantity of products, cooking techniques, and presentation based on preferences and tolerances. The reintroduction of previously eliminated foods should start with small amounts and gradually increase to well-tolerated quantities.The results of studies conducted so far indicate promising outcomes of probiotic supplementation, psyllium, and vitamin D. However, further research is needed to definitively confirm their effectiveness.Regular physical activity is a crucial element in supporting IBS therapy. Physical exertion positively affects overall health, bodily functions, well-being, and mood. It may also provide benefits in terms of symptom severity and frequency. The most commonly recommended forms of physical activity among individuals with IBS include walking, cycling, swimming, yoga, or aerobics.

## Figures and Tables

**Figure 1 nutrients-15-03662-f001:**
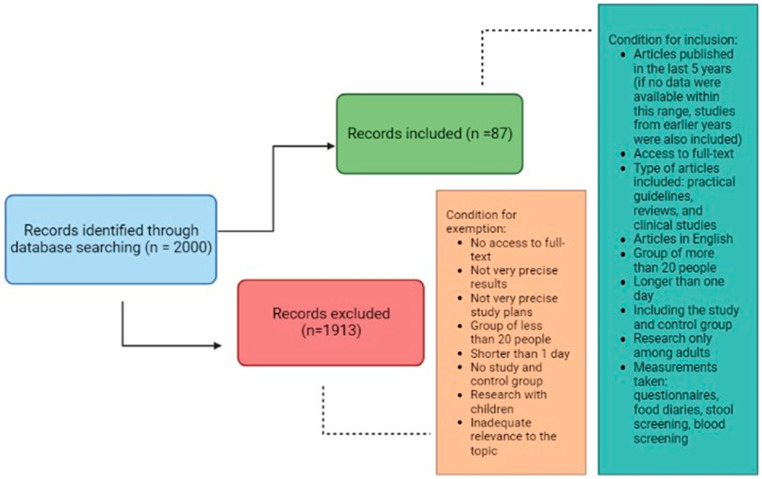
The conditions for inclusion and exclusion of articles.

**Figure 2 nutrients-15-03662-f002:**
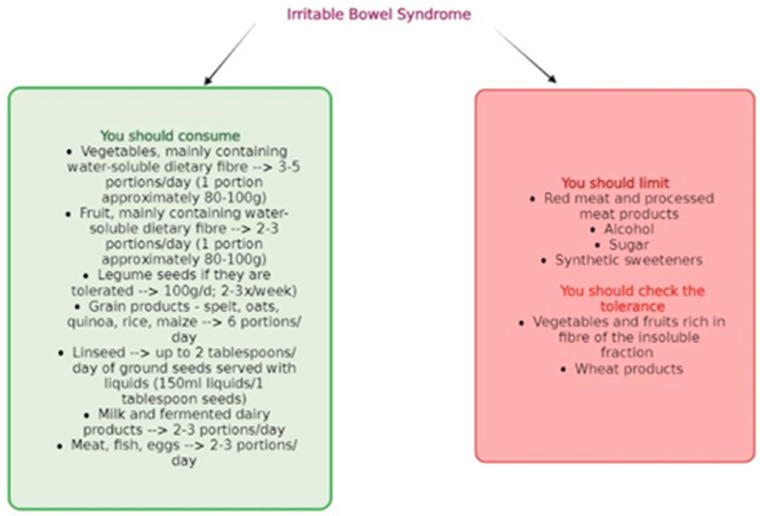
General dietary recommendations for patients with IBS.

**Table 1 nutrients-15-03662-t001:** Products rich in specific types of FODMAP [21].

Types FODMAP	The Main Products Rich in FODMAP
Fructans (fructooligosaccharides, inulin, oligofructose)	Wheat, vegetables (including artichokes, onion-la, garlic, cauliflower, asparagus, broccoli, mushrooms)
Galactooligosaccharides	Legumes, lentils, chickpeas (hummus)
Disaccharides (lactose)	Milk and milk products
Monosaccharides (fructose and high fructose/high glucose)	Honey, corn syrup, fruit juices in large quantities (pineapple, watermelon, pear, apple), fruit (including apples, mangoes, figs, watermelon, grapes)
Polyols (sorbitol, mannitol, xylitol, isomalt)	Dried prunes, apples, mushrooms, avocados, cauliflower, sweeteners for sugar-free products, including puddings, gelatine, chewing gum, mints, sweets

**Table 2 nutrients-15-03662-t002:** Clinical studies evaluating the effectiveness of dietary interventions in the course of IBS.

Country, Study, Year	Design, Population	Interventions	Main Findings
Australia, Biesiekierski et al., 2013 [24]	1 stage: randomized, double-blind, placebo-controlled, cross-over trial, patients with IBS and non-celiac gluten sensitivity (*n* = 37) 2 stage: rechallenge, patients with IBS and non-celiac gluten sensitivity (*n* = 22)	1 stage: High gluten (16 g/day) vs. Low gluten (2 g/day) vs. Whey (16 g/day), 1 week per intervention. 2 stage: Gluten (16 g/day) vs. Whey (16 g/day) vs. Placebo (no additional protein), 3 days. Run-in period of 2 weeks, gluten-free diet and low FODMAP diet	During reduction of FODMAP intake: improvement of gastrointestinal complaints in all subjects Diet containing gluten and whey protein: significant worsening of symptoms No evidence of a specific and dose-dependent response to gluten
Norway, Skodje et al., 2018 [23]	randomized, double-blind, placebo-controlled, cross-over trial, self-reported patients with non-celiac gluten sensitivity on gluten-free diet >6 months (*n* = 59)	Gluten-free diet (placebo-concealed muesli bars) vs. gluten-containing diet (5.7 g/day) vs. fructans containing diet (2.1 g/day), 7 days per intervention, 7 days washout	Significant differences in gastrointestinal symptoms between different dietary interventions Greatest symptoms among those consuming fructans No significant differences in symptoms between the placebo group and those consuming gluten
UK, Parker et al., 2001 [25]	No randomized controlled trial, observational interventional study 1 stage: patients with IBS were given lactose hydrogen breath tests (*n* = 122) 2 stage: patients with IBS and positive lactose hydrogen breath tests (*n* = 23) 3 stage: double-blind, placebo-controlled challenges, patients with IBS and positive lactose hydrogen breath tests and improving on the diet to confirm lactose intolerance (*n* = 9) 4 stage: patients who did not respond to the low lactose diet (*n* = 9) 5 stage: patients with IBS and negative lactose hydrogen breath tests (*n* = 35) Assessment of symptoms: before lactose hydrogen breath tests, 8 h after lactose hydrogen breath tests, every day during each dietary change	2 stage: low lactose diet for 3 weeks 3 stage: diet containing 5 g vs. 10 g vs. 15 g of lactose vs. placebo 4 stage: followed either an exclusion or low fibre diet 5 stage: other dietary interventions	Before lactose hydrogen breath tests: no significant differences in symptoms After lactose hydrogen breath tests: symptoms in the positive group significantly worse Low lactose diet: improvement in 39% of people among those following the diet Exclusion diet: improvement in 50% of people among those following the diet Low fibre diet: improvement in 2/3 of those following the diet Lactose-free diet has no benefit among people with IBS regardless of test result
Netherlands, Bohmer et al., 2001 [26]	No randomized controlled trial, prospective observational study, patients with IBS and lactose malabsorption (*n* = 17) vs. patients with IBS and lactose tolerance (*n* = 53)	Low lactose diet and assessment of symptoms before, during, 6 weeks after and 5 years after starting the diet	Before lactose hydrogen breath tests: no significant differences in symptoms 6 weeks after starting diet: significant improvement in people with lactose malabsorption 5 years after starting the diet: significant improvement in people with lactose malabsorption Among people with IBS, it is very important to perform a lactose tolerance test and to include a lactose-free diet among those with a positive test result
Netherlands, Bijkerk et al., 2009 [27]	Randomized Controlled Trial, patients with IBS (*n* = 275) Observation of an increase in dietary fiber of the soluble (psyllium) or insoluble (bran) fraction in the diet	12 weeks diet containing 10 g psyllium (*n* = 85) vs. 10 g bran (*n* = 97) vs. 10 g placebo (rice flour) (*n* = 93)	After 4 weeks and 2 months, a significant improvement in symptoms was noted among the psyllium group compared to the placebo group No significant effect of bran on symptoms compared to placebo After 12 weeks, a significant improvement in symptoms was noted among the psyllium group compared to placebo and the bran group
Sweden, Bohn et al., 2015 [28]	Multicenter Randomized Controlled Trial, patients with IBS (*n* = 75) Evaluation before and after intervention	4 weeks traditional IBS diet (NICE guidelines) vs. Low FODMAP diet	During dietary intervention: relief of discomfort in both groups, with no significant difference between groups After 4 weeks of dietary intervention: 50% of those following the low FODMAP diet reported symptom relief vs. 46% of those following NICE recommendations reported symptom relief
Australia, Halmos et al., 2014 [29]	Single-centre, Randomized Controlled Trial, cross-over, patients with IBS (*n* = 30) vs. healthy control (*n* = 8) Evaluation before, during and after intervention	21 days low FODMAP diet vs. typical Australian diet with a washout period of at least 21 days	During the diet: overall gastrointestinal symptoms were significantly reduced in the group on the low FODMAP diet compared to the control group. Flatulence, abdominal pain and gas also eased in the low FODMAP group. Reported stool consistency significantly better on the low FODMAP diet
Australia, Halmos et al., 2015 [30]	Single-blinded, randomised, cross-over trial, patients with IBS (*n* = 27) vs. healthy control (*n* = 6), Evaluation before, during and after intervention	21 days low FODMAP diet vs. typical Australian diet with a washout period of at least 21 days	The low FODMAP diet group had higher stool pH, similar concentrations of short-chain fatty acids, and higher microbial diversity and reduced total bacteria compared to the control group. Low FODMAP diet significantly affects gut microbiota composition in the short term, long-term studies needed

**Table 3 nutrients-15-03662-t003:** Types of diets used in IBS and their effectiveness.

Type of Diet	Dietary Assumptions	Effects on IBS Based on Research
Gluten-free diet	Elimination of gluten, i.e., products containing wheat, barley, rye, oats and related grains. It is recommended to eat, among other things, fruit, vegetables, fish, meat and gluten-free products [30].	Research has shown that components of wheat may be responsible for causing some of the symptoms of IBS. However, there is no evidence that gluten is a factor. Therefore, a gluten-free diet should not be recommended as standard for people with IBS and more research is needed to assess the effect of gluten on IBS [17,23,31].
Lactose-free diet	Limit consumption of lactose to 12 g/day. Eliminate the consumption of milk (cow, goat, sheep) and dairy products [33].	Tests for lactose intolerance should be performed among patients with IBS. However, this diet should not be recommended to all people with IBS [17,24,32].
High-fiber diet	Increasing the intake of foods rich in fibre of the water-soluble fraction and introducing additional amounts in the form of Psyllium seed husks [17].	Strong research. Dietary fibre supplementation of the water-soluble fraction (e.g., Psyllium husks), may have a beneficial effect on the course of IBS [17,25].
Low FODMAP diet	Elimination of products rich in fructans, galatooligosaccharides, disaccharides, monosaccharides and polyols [21].	Low-quality research. Short-term use of the diet, may be beneficial in relieving symptoms. However, prolonged use reduces the diversity of the gut microbiota. Therefore, once IBS symptoms have abated, the diet should be expanded according to tolerance. More studies are needed to confirm efficacy [17,26,27,28].
NICE guidelines	General recommendations such as eating regularly, avoiding skipping meals and large meals, drinking approximately 2 litres of fluids/day, limiting the consumption of alcoholic and carbonated beverages, reducing the intake of caffeine, fat, dietary fibre of the insoluble fraction, resistant starch and gas-enhancing products [17,19].	Current recommendations given to every patient with IBS. The most effective and safe nutritional intervention [19,29].

## Data Availability

Not applicable.

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
