# Peer review of "Nutrition, Physical Activity and Supplementation in Irritable Bowel Syndrome"

_nutrients, 2023, doi:10.3390/nu15163662_

Round 1
Reviewer 1 Report
The review provides an overview of Irritable Bowel Syndrome (IBS) as a chronic functional disorder of the intestines. It emphasizes that individuals with IBS often associate the severity of their symptoms with the food they consume, leading to dietary restrictions and a search for appropriate dietary selections.
In this review authors discussed supplementation as a supportive element in IBS therapy. While components are mentioned, there is no in-depth discussion of their mechanisms of action or specific evidence supporting their effectiveness in managing IBS symptoms.
In methods section, authors mentioned they have found 2000 literature items, how many are selected what was the selection criteria. This is not a systemic review, however, it would add and advantage to this review and for understanding the data availability on selected components linking with IBS.
Reviewer 2 Report
This review article aimed to analyse the existing literature dealing with the impact of different food products, physical activity, and selected supplementation on the development of Irritable Bowel Syndrome (IBS). Actually, there are still several uncertainties to be clarified as well as etiopathogenesis, physiopathology, diagnostic criteria and therapy of IBS are concerned.
Introduction disserts with full knowledge on a series of aspects concerning IBS (i.e. clinical subtypes, epidemiology, multifactorial genesis, systemic fall-out effects including anxiety and depression, symptomatic pharmacological treatments, dietary incongruities, and reduced physical activity.
Materials and Methods indicate the criteria that were employed to set out relevant studies from the PubMed database (articles published within the last 5 years were predominantly considered).
Eating habits were examined including FODMAP diet and other types of diets used in IBS with their effectiveness. Food groups were then analysed with reference to IBS (i.e. vegetables and fruit, legume seeds, grain products, milk and dairy products, and meat-fish-eggs). Figure 1 fairly resumes general dietary recommendations for patients with IBS. Afterwards, Authors devote particular attention to the problem of supplementation in IBS and, in this respect, they discuss the question of the employment of probiotics and Psyllium husk also with relation to the gut microbiota. Moreover, Authors underline the role of vitamin D, which is thought to be very important for IBS management, and recommend and adequate and regular physical activity. In conclusion, the dietary approach in IBS should correspond to the particular patient’s condition as resulting from both course and symptoms of the disease.
Overall, this review has been well planned and well written representing an efficacious synopsis of how nutrition, physical activity and supplementation are able to affect the course of IBS. The manuscript has been prepared with care and corroborated by faithful bibliographical references.
Lexicon, sentence fluency, “English style” and expository fashion are adequate. Here and there, only some minor editorial refinements might be appropriate (e.g. line 9:…intestines…=…intestine).
Reviewer 3 Report
This paper intends to review the literature on effective interventions regarding Irritable Bowel Syndrome like nutrition, dietary supplementation and physical activity. However, the review is poorly executed and does not meet the standards for systematic nor scientific reviews.
Some examples that support the lack of standards:
Lines 99/100 It is needed to specify the inclusion criteria for papers of ‘earlier years’.
Lines 110/113: define imprecise results, study plans or inadequate relevance to the topic.
For a systematic review the inclusion and exclusion criteria are poorly described. Lacking is a flow diagram. And only searching through PubMED is not sufficient. Why are the specific keywords used? Please describe.
Figure 1. General dietary recommendations for patients with IBS, still has the tag ‘Created in BioRender.com’.
The results start with Section 3. Eating habits, Line 114? Actually, Lines 115-131 + 134-140 describe the current recommended eating habits/patterns and diets per the authority. Two Tables with results of the systematic review are shown. Table 1: Products rich in specific types of FODMAP. Is this a result of the systematic review? Why is it here, as more specific food groups are addressed in the following sections. So maybe this should be moved to a discussion, when giving guidelines? Table 2 includes references 17, 19, 21, 23, 24 of which most are not original research, but reviews in itself. Besides, these limited studies, of which surely more medical trials are available, are not even well presented by mentioning the study, intervention, diet characteristics, study participants, measured indicators, and the actual study outcomes. For all the sections 4 through 6, the text is abundant, but is lacking the support of Tables that summarize the study parameters.
Lines 40/42 should read: ‘ If all these criteria are met and there are no additional alarming symptoms, there is no need for further diagnostic tests and the diagnoses IBS can be set [1], with further characterisatin according to the Bristol Stool Form Scale that classifies IBS into four subtypes: ….’
Lines 47/52 should read: ‘The occurrence of IBS is characterized by significant variability both between countries and within a single country [7], affecting about 7-21% of the general population, most commonly reported in South American countries [3,4,7,8]. IBS is diagnosed twice as often in women compared to men [8], and lthough the condition can be diagnosed at any age, it is most commonly diagnosed in women before the age of 50 [4,7].’
Lines 53/56 should read: ‘The etiopathogenesis of IBS is multifactorial and currently not fully understood [4]. According to existing analyses, it is possible that IBS results from interactions between genetic factors, psychiatric disorders, dysregulation of the hypothalamic-pituitary-adrenal axis, visceral hypersensitivity, and inflammatory states within the gastrointestinal tract as well as other parts of the body [1,3,5].’
Line 58 and 72 and 76 and 96 replace with ‘Irritable Bowel Syndrome‘ with ‘IBS’
Line 58/65 resposition to link with the subtypes?
Line 72, new paragraph starting with: ‘However, the 72 greatest value…’ and merge with ‘Even up to 70% of individuals…’
Line 80, replace :’their’ with ‘the quality of life of individuals with IBS’ or ‘IBS patients’’
Line 80 start new paragraph with, and rewrite: ‘In addition to dietary changes, important modifiable factors for individuals with the condition that can influence the course of IBS are dietary supplementation and physical activity. Dietary supplements, although their effects on the human body are not precisely proven {references needed}, may have beneficial effects on human health. Individuals with IBS can benefit from additional dietary supplementation in terms of reducing symptoms, improving well-being, and enhancing quality of life [13].
Line 86, start new paragraph: ‘Due to the accompanying symptoms of the disease, patients often limit their engagement in physical activity…. course of the disease among individuals with IBS [14,15,16].
Line 91 start new paragraph and reformulate the purpose of the review.
For example:
Due to the limited evidence regarding the impact of different food products, physical activity, and selected dietary supplementation on the course of IBS, unconclusive guidelines are available, and both patients and professional medical personnel lack reliable information regarding interventions that encompass rational dietary modifications, physical activity, and the inclusion of appropriate dietary supplements. Therefore, the aim of this study is to review the existing literature that describe the interventions effective for IBS.
2. Methods
Lines 99/100 should read: ‘A systematic literature search was conducted using the PubMed database to identify studies relevant for the current review with the limitation to articles published within the last 5 years; however, if no data were available within this range, studies from earlier years were also included.’
Lines 99/100 It is needed to specify the inclusion criteria of ‘earlier years’.
Lines 110/113: define imprecise results, study plans or inadequate relevance to the topic.
Round 2
Reviewer 3 Report
There are many statements in this manuscript that do not differ from other recent reviews, while also not completely based on proven and repeated clinical trials. For example, lines 231/232: ‘Individuals with IBS should aim to consume 3-5 servings of vegetables per day and 2-3 servings of fruits per day [13].’ Reference 13 is a review and the claim is not specifically based on clinical evidence from an RCT. Nevertheless, I think the manuscript is worth its discussion and its approach. Even though I think the systematic approach to search for papers is not applied properly. However, I would like to request some improvements and suggest describing the search with inclusion and exclusion criteria in more detail (under methods), and include more text (and rearrange) the section on the tabulated clinical trials and diets (section 3), before going into reviewing specifics like 4 Food groups, 5 Supplements, 6 Physical activity.
Following are the more detailed instructions for improvement:
Write Irritable Bowel Syndrome in full only once (introduction), and then stick to the abbreviation IBS.
Line 42/43 and 50: I previously prepared suggestions with typos. Please use ‘characterization’ and ‘although’.
My previous edit ‘however, if no data were available within this range, studies from earlier years were also included’ was merely a suggestion. Now that I read the reformulated aim, and the more detailed inclusion and exclusion criteria (mostly extracted from my suggested edits), I am still skeptical whether the manuscript actually follows PRISMA.
Fist the inclusion and exclusion criteria as mentioned in Figure 1. Full-text availability nowadays can hardly be a scientific criterion. In general, it is possible to obtain a full text PDF from the author if the title and abstract are interesting enough, especially through personal communication e.g., by means of ResearchGate, Academia, LinkedIn. Language limited to English could be a criterium.
The exclusion criteria: ‘Not very precise results’ and ‘Not very precise study plans’ are clearly indicating opposite criteria were applied that requested a minimal description of the results and study plan, however, these are not given in the inclusion criteria. Intervention duration, comparing various intervention groups, participants, measurements performed (questionnaires, surveys), study outcomes. I guess a reason for doing this review is to identify the most recent clinical trials/interventions with proper groups and diets relevant to IBS. That requires minimal participants to reach a statistical difference based on methodology, indicators, and outcomes. Case studies of describing only a few participants are therefore not relevant (and could be an exclusion criterion).
‘Inadequate relevance to the topic’ is vague, as the extensive search string should provide relevant papers.
Thus, please provide more detail on the inclusion and exclusion criteria.
Second, there was a reason to ask for tabulation of the studies selected, and the request to include intervention, diet characteristics, study participants, measured indicators, and the actual study outcomes. While the newly prepared Table is a very good effort, this review should summarize which of these trials was effective, or has the potential to be effective. Thus Table 3 should be discussed in more detail as to make a distinction from other reviews on IBS. Now the section with lines 141-161 is short and non-conclusive, giving dietary professionals no direct lead of what diet is effective (and sure it can be a conclusion one diet does not work for all IBS patients). Only 8 clinical trials are mentioned in Table 3, of which the most recent is from 2018, and others are 8 years old already. Perhaps a better presentation would be: first the clinical trials with table 3, discuss the studies and the results, and then write the general conclusion on diets (lines 141-146) with table 2 types of diets. I mean it is kind of strange to first state ‘the low FODMAP diet currently has the most extensive evidence for its effectiveness in dietary therapy among patients with IBS’ and later write ‘However, the number of studies conducted to date is insufficient to unequivocally recommend a single diet’.
So, please restructure this important first section 3 Eating habits
Line 121: ‘Eating Habits’ to be replaced by ‘Food Choices’?
The lines 158-160 ‘In addition, these diets may expose those following them to the elimination of valuable nutrients and the possibility of deficiencies [17,21,23,25].’ could be used introduce the next part of the manuscript (section 4-6) and to state that the remaining of the manuscript will explain and discuss that diets for IBS patients could come with deficiencies, and that specific iterative food choices for individual patients would be best to establish and consider more integrative diet and food group choices based on personal effectiveness.
